# How improving access times had unforeseen consequences: a case study in a Dutch hospital

Oskar Roemeling,[1] Kees Ahaus,[2] Folkert van Zanten,[3] Martin Land,[3] Patrick Wennekes[4]

[1]Innovation Management & Strategy, University of Groningen Faculty of Economics and Business, Groningen, The Netherlands
[2]Health Services Management & Organization, Erasmus University Rotterdam Institute of Health Policy and Management, Rotterdam, The Netherlands
[3]Operations, University of Groningen Faculty of Economics and Business, Groningen, The Netherlands
[4]Process Management and Improvement, Martini Hospital, Groningen, The Netherlands

**Correspondence to**
Dr Oskar Roemeling;
o.p.roemeling@rug.nl

## ABSTRACT

**Objectives** To investigate the consequences of increasing capacity to reduce access times, and to explore how patient waiting times and use of physical capacity were influenced by variability.

**Design** A retrospective case study that combines both primary and secondary data. Secondary data were retrieved from a hospital database to establish inflow and outflow of patients, utilisation of resources and available capacity, realised access times and the weekly number of new patients seen over 1 year. Primary data consisted of field notes, onsite visits and observations, and semistructured interviews.

**Setting** A secondary care facility, that is, a rheumatology department, in a large Dutch hospital.

**Participants** Analyses are based on secondary patient data from the hospital database, and the responses of the interviews with physicians, nurses and Lean Six Sigma project leaders.

**Results** The study shows that artificial variability was increased by managerial decisions to add capacity and to allow an increased inflow of new patients. This, in turn, resulted in undesirable and significant fluctuations in access times. We argue that we witnessed a new multiplier effect that typifies the fluctuations.

**Conclusions** Adding capacity resources to reduce access times might appear an obvious and effective solution. However, the outcomes were less straightforward than expected, and even led to new artificial variability. The study reveals a phenomenon that is specific to service environments, and especially healthcare, and has detrimental consequences for access times.

## Strengths and limitations of this study

► The study provides knowledge as to the effectiveness of increasing capacity buffers to reduce hospital access times.

► A particular strength of this study relates to the novel structured approach adopted to investigate hospital access times that generates a thorough understanding of actions to reduce these and their consequences over a prolonged period.

► Another strength of this study is its combination of both objective performance data retrieved from the hospital database and subjective qualitative data to explain past occurrences.

► Although the dyadic structure of general practitioners and a small rheumatology department with new and returning patients is relatively simple and open to analysis, it does not represent the more complex networks with multiple interdependencies seen in many other situations.

► Another limitation relates to the adopted single case study approach which limits the generalisability of the findings, although the phenomena we identify are widely relevant and not unique to the setting.

help understand the underlying mechanism that causes the adverse effects.

Patient flow is a topic of ongoing interest in healthcare research.[1–4] However, obtaining flow is not easy, and one has to be sensitive to barriers such as limited internal integration,[5] paradoxes such as improvements that target parts of a system, but fail to address underlying system constraints,[6] and strategies that do not address the linkages between process, population and capacity.[7] Especially, limited capacity can be a considerable bottleneck that impedes patient flow. Hence, it makes sense to increase capacity to alleviate this process constraint.[8] In turn, a smooth patient flow is obtained, characterised by short patient waiting times.

In our study, we are especially interested in the access time as a specific type of waiting time. Achieving short access times is important

## INTRODUCTION

It seems very logical that, when confronted with long access times, one increases capacity to more quickly serve clients: an obvious solution to a clear problem, or so it appears. However, this study shows the possible adverse consequences of adding physical capacity to reduce access times, which we define as the time between referral from the general practitioner (GP) to the first visit of the patient to the rheumatology department. In addition, we provide a theoretical frame of reference to

for healthcare providers: reducing access times can positively affect the patients' condition (eg, by reducing pain) and can prevent further problems as medical conditions can deteriorate while waiting for treatment.[9 10] Despite this, consistently achieving short access times is difficult. One of the main barriers to achieving short access times, and short waiting times in general, is variability.[11]

One can distinguish between natural and artificial variabilities.[12–14] Natural variability is inevitable in healthcare and can only be controlled to a very limited extent: natural variability is a 'fact of life', for example the different responses to similar treatment between patients. In contrast, artificial variability is caused by our own actions, routines or regulations. For example, variability stemming from providing inconsistent quality of referral information by GPs.[15] This type of variability can and should be controlled and reduced. (For a more elaborate discussion of artificial variability and related examples, see Dempsey[16])

The mechanisms that allow an organisation, or more precisely a process, to cope with variability are termed buffers. Some form of buffer will be required to cope with variability in either supply or demand.[17] The available buffers in a healthcare environment consist of time buffers (waiting patients) and capacity buffers (idle resource capacity).[18] Healthcare is thus faced with a trade-off between either waiting patients or idle capacity, both of which are unattractive. Buffers are the natural result of a system that has to cope with variability, but one should continuously attempt to minimise buffers and reduce their underlying causes.[19]

Increasing capacity buffers seem to be the most effective solution if one wants to reduce the time buffers represented by hospital access times. However, during a recent improvement project, we witnessed adverse outcomes from expanding capacity to cope with long access times. This led us to the following research question: why does expanding capacity lead to variation in patient waiting times? This study aims to contribute to knowledge on the dynamic relationship between variability and patient waiting times, and to the understanding of fluctuations in hospital access times.

In the remainder of this paper, we first introduce the methodology applied by explaining the hospital setting and the approaches used in obtaining and analysing the empirical data. Following this, we present the results of our analysis, and then a discussion reflecting on the interaction between variability and buffers. This interaction turned out to be less straightforward than expected. This paper then ends with the conclusions in which we highlight our main contributions.

## METHODS

The research setting for this study was the rheumatology department of a Dutch teaching hospital. The hospital had embarked on an improvement programme with several projects focussing on improving flow performance. These projects allowed us to study the impact of actions aimed at reducing access times defined as the time from the referral of the GP to the first visit of the rheumatology department. The rheumatology department employed two specialists (physicians), a specialised nurse practitioner and a rheumatology patient advisor that together served 9277 patients in 2013. As the access time was rather long (up to 16 weeks) and the utilisation rate of the two physicians was high, the hospital board decided to initiate an improvement project.

The aim of this research was to understand the complex interaction between variability and buffers. To study a complex phenomenon in its natural setting, we applied the single retrospective case study methodology.[20] While our study was quantitative in nature, we have added a qualitative component to explain past events, and strengthen our understanding of the identified phenomenon. The data gathered were used to study the interaction between variability and buffering and the fluctuations in access times through an operations management lens, a perspective especially appropriate for issues related to patient flow.

## Case description

The typical patient pathway in our case setting is as follows. A patient experiences problems with their joints, and visits their GP. The GP will refer the patient to a rheumatologist, who will examine the patient at an outpatient clinic. If deemed necessary, the rheumatologist will request additional diagnostic tests (an ultrasound examination, X-ray, blood test, etc). The rheumatologist will then discuss a treatment plan with the patient and notify their GP. Alongside the rheumatologists, the nurse practitioner serves a small, predefined, group of new patients whose symptoms lack complexity.

The rheumatology department distinguishes three types of consultation requests determined by the perceived urgency of the problem: (1) regular, (2) semiurgent and (3) urgent patients. Of the new patient arrivals, 7% are considered urgent and 6% semiurgent. Standard access times have been predetermined for urgent and semiurgent patients (2 weeks and 4 weeks respectively). No target access time has been set for 'regular' patients who form the majority of new arrivals. Before the improvement project was initiated, the average access time across patients was around 10 weeks with peaks up to 16 weeks.

## Data sources

This study combines several secondary data sources from the hospital database: inflow and outflow (new arrivals and patients leaving the system), utilisation of resources and available capacity, realised access times and the number of new patients per week over 1 year (2013). In addition to these secondary data sources, we acquired primary qualitative data through keeping field notes, onsite visits and observations, and semistructured interviews with physicians, nurses and Lean Six Sigma (LSS) project leaders.

During the interviews, questions were focused on obtaining in-depth information on possible sources of variability. We used an interview protocol to enhance

both the reliability and the validity of the research.[20] Combining objective quantitative and subjective qualitative data contributes to the accuracy and reliability of case study research.[20] In addition, the physicians, nurses and LSS project leaders had a crucial role during a focus group meeting in which we reflected on the outcomes of the analyses and attempted to build an understanding what had occurred.

## Analysis approach

This study required an in-depth exploration of access time data, for which there is no standard approach. Therefore, the initial analysis involved calculating the median and mean access times for each week. The median is often used in analyses of highly variable data because it is less sensitive to outliers than the mean. In addition, we performed an independent sample t-test on the means of the access times to compare the pre-intervention and post-intervention periods.

Next to median access times, we calculated the 90th percentile of the access times for each week to detect whether any apparent changes in access times affected a large majority (ie, at least 90%) of patients. We constructed diagrams that relate capacity and access times over time, through which we provide insights into the dynamics of time and capacity buffers.

To deepen the analysis of flow-related variables, we followed a previously used approach[21] making use of throughput diagrams that show both cumulative patient inflow and outflow curves over time. The vertical distance between the two curves reflects the number of waiting patients while the horizontal distance reflects the backlog in terms of time. The behaviour of these two variables can be analysed over time and at the same time be related to fluctuating patient inflow and outflow rates. As such it is particularly useful in dynamic situations where the averages change over time. Finally, the relative share of new and returning patients was analysed to detect any ratio changes over time.

In the analyses, the weekly capacity in person-days is calculated based on the number of scheduled physicians and corrected for any absences. The capacity information is based on the available calendars of the physicians. This capacity is allocated to new and returning patients.

## Patient and public involvement statement

This research did not involve patients in the design, conduct and reporting of the research. The results of this work were presented and discussed during a focus group meeting with representatives of the hospital. Once the study has been published, the main findings will be used in the education of Master students.

## RESULTS

Figure 1 depicts the capacity in person-days each week and the access times (in weeks) for patients being referred by their GP in that week. The analysis shows that, up to week 31, the maximum capacity was 10 person-days, that is, two full-time physicians.

In an attempt to reduce the access times, an intern (shown in red) provided additional capacity from week 31.

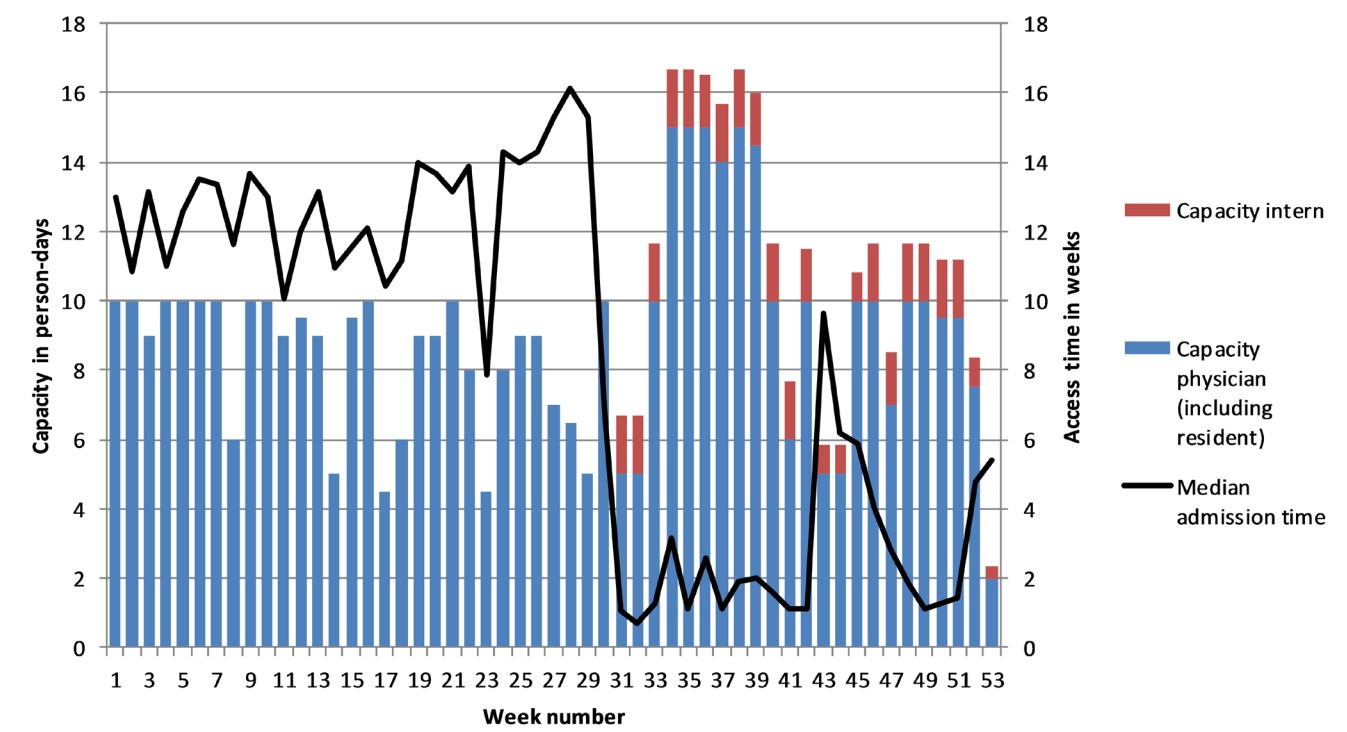

**Figure 1** Weekly capacities in person-days and median access times.

In addition, in weeks 33 through 39, a resident provided a further temporary boost to capacity. Here, we note that the productivity of the resident was expected to be similar to that of the rheumatologists, whereas the intern would achieve roughly one-third of their output due to a lack of experience. The graph is adjusted to reflect this difference. In effect, by increasing staff, the department sought to replace time buffers (waiting patients) with capacity buffers (additional physicians).

The expansion in capacity resulted in an almost immediate drop in access times from week 31 onwards. The reduction of access times, measured in days, for individual patients is significant at a level of p<0.001 when comparing patients that arrived in weeks 1–29 (M=72, SD=42) with patients that arrived in weeks 32–52 (M=29, SD=38). Weeks 30 and 31 have been excluded as the transition period. In terms of variability, figure 1 shows that the actual capacity in just over half of the weeks of the first 6 months was below the notional 10 person-days, and often significantly so. That is, there were more weeks in which at least one rheumatologist was partly absent than weeks in which both were fully present. This illustrates that, in this case, capacity availability is a relevant source of artificial variability.

Furthermore, figure 1 shows that, in week 28, more than half of the appointments made had an access time of at least 16 weeks in the future (the median value). Whereas, in week 32, half of the appointments made were for within 1 week. In other words, patients that sought an

appointment in week 28 often had to wait a further 16 weeks for their meeting with a physician, whereas patients that joined the list in week 32 would usually see their physician within a few days.

When we look at the 90th percentile of recorded access times (figure 2), there is no equivalent and sustained immediate improvement after week 31, indicating that a substantial number of patients still experienced lengthy access times. However, from week 46 on, there was a significant drop in the 90th percentile, indicating that the access times for the majority of patients had decreased in the final weeks of the year.

Influences on access times can also be analysed using throughput diagrams (figure 3). The vertical axis shows cumulative numbers of patients. The upper (green) curve represents the cumulative inflow and the lower (blue) curve the cumulative outflow. To ensure data validity, the inflow curve is not shown after week 47 because some of the patients then applying for appointments might not have access until the first weeks of the following year, which would distort any conclusions.

The horizontal distance between the two curves in figure 3 is the backlog (in weeks) and can be interpreted as the projected access time for a patient requesting an appointment at that point in time. A red curve, representing this horizontal distance, is included, with values indicated on the secondary vertical axis. For example, the 400th request for an appointment (inflow) was made in week 17, and the 400th appointment (outflow) was

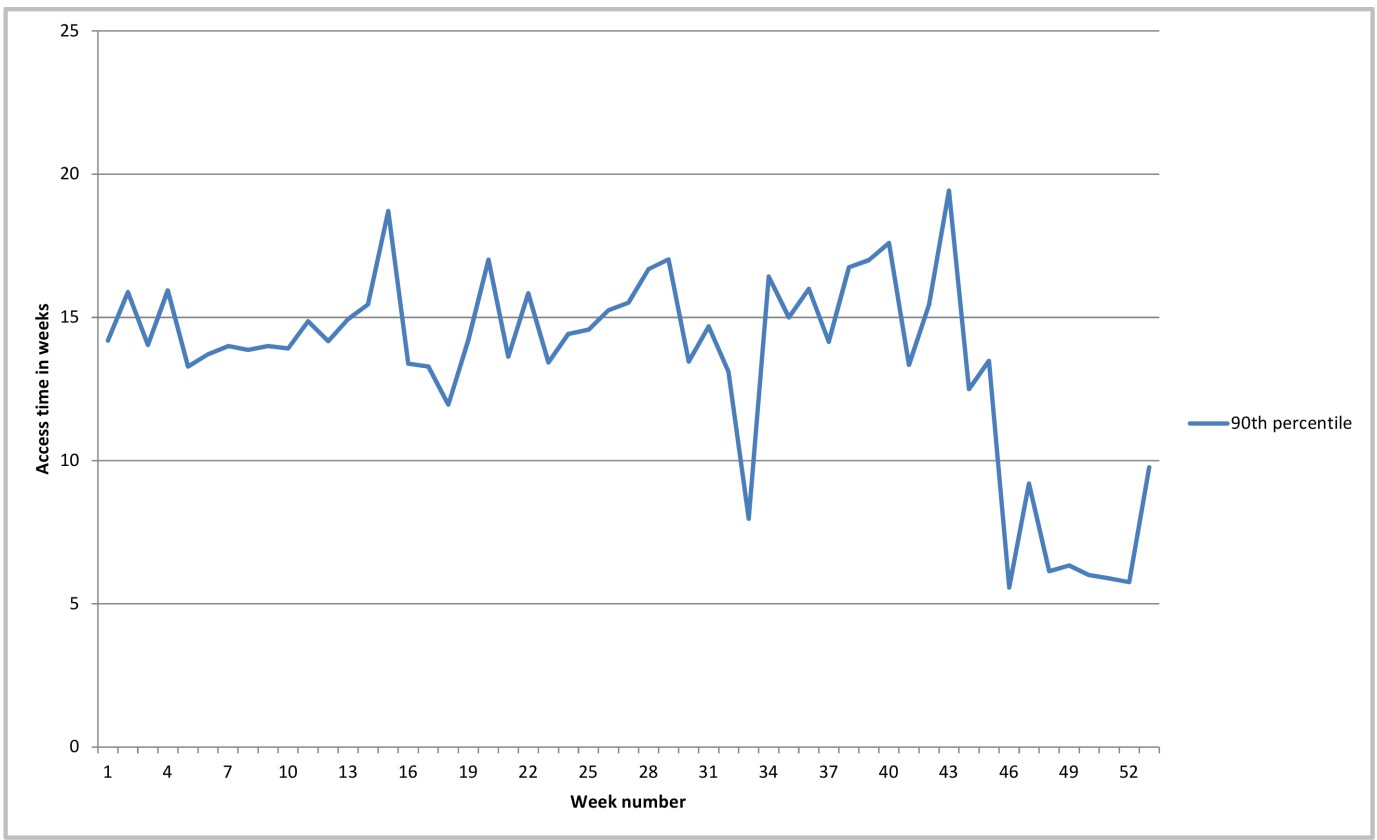

**Figure 2**  90th percentile of access times.

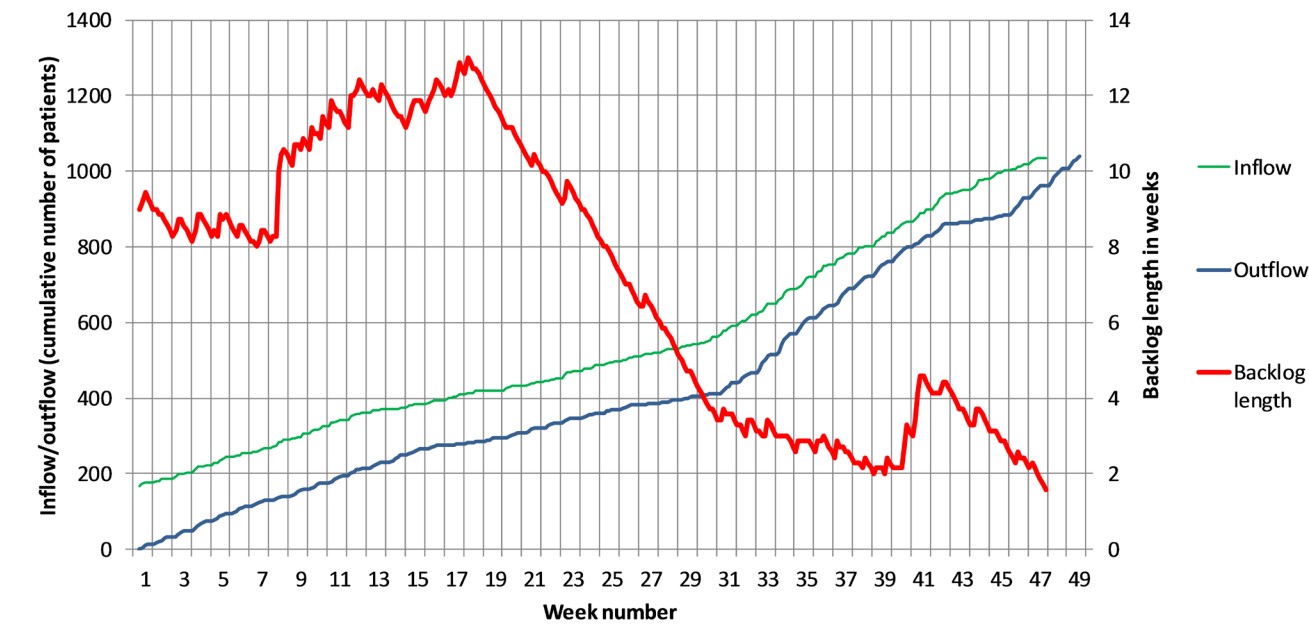

**Figure 3** Throughput diagram including derived mean access times.

scheduled for week 29 (but not necessarily the same patient). This means that a patient joining the queue in week 17 could, on average, expect an access time of 12 weeks, whereas by the 800th patient (arriving in week 38) the expected access time was down to 2 weeks. Compared with the representations in figures 1 and 2, this diagram has the advantage of being less sensitive to individual deviations from the typical access times. It provides a global view of how access times develop and how they can be related to surges or drops in patient inflow or outflow. More specifically, figure 3 shows how the decrease in mean access times from week 31 onwards was indeed

linked to a surge in outflow, rather than a drop in inflow. As such, it was clearly the result of the capacity increase.

Along with the sudden decrease in access times due to the increase in capacity, the number of new patients being referred by their GPs increased significantly. For example, figure 4 shows an increase from 10 new referrals in weeks 29 to 30 in week 31.

From the interviews, we learnt that referrers (ie, GPs) select those hospitals that have the shortest access times. An unforeseen consequence of this was that the drastically reduced access times in the studied unit had a pulling or magnetic effect. Once the extra capacity was added, extra

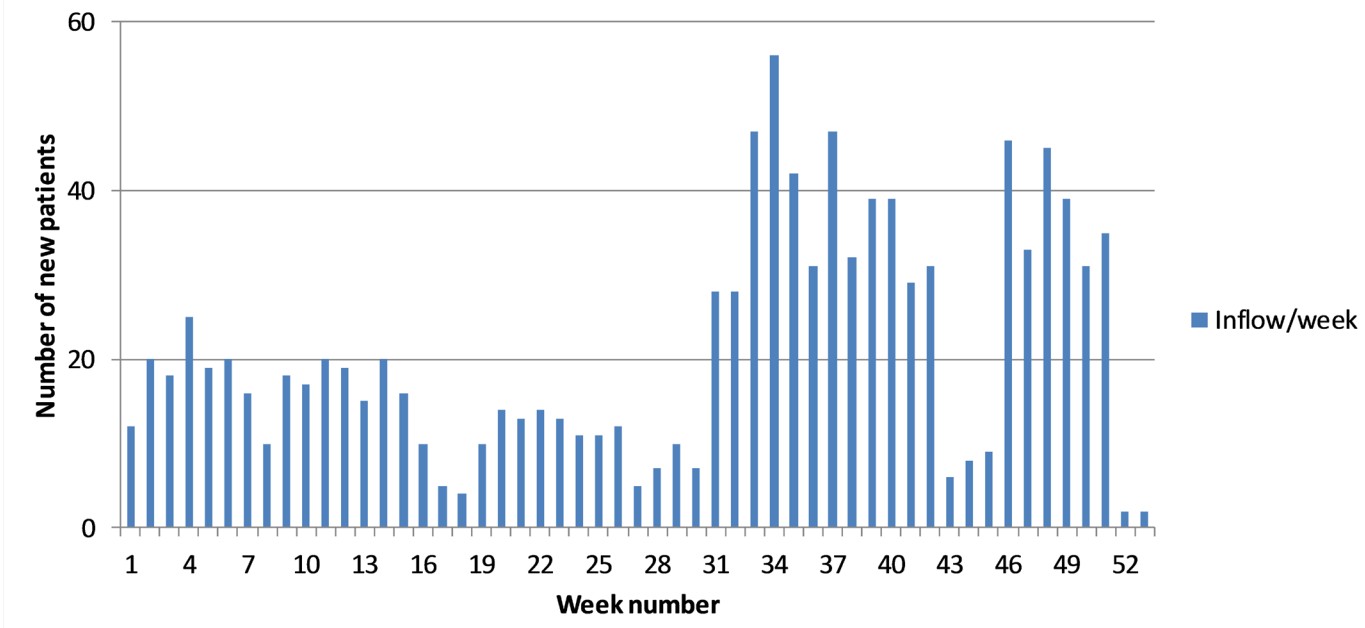

**Figure 4** Weekly inflow of new patients.

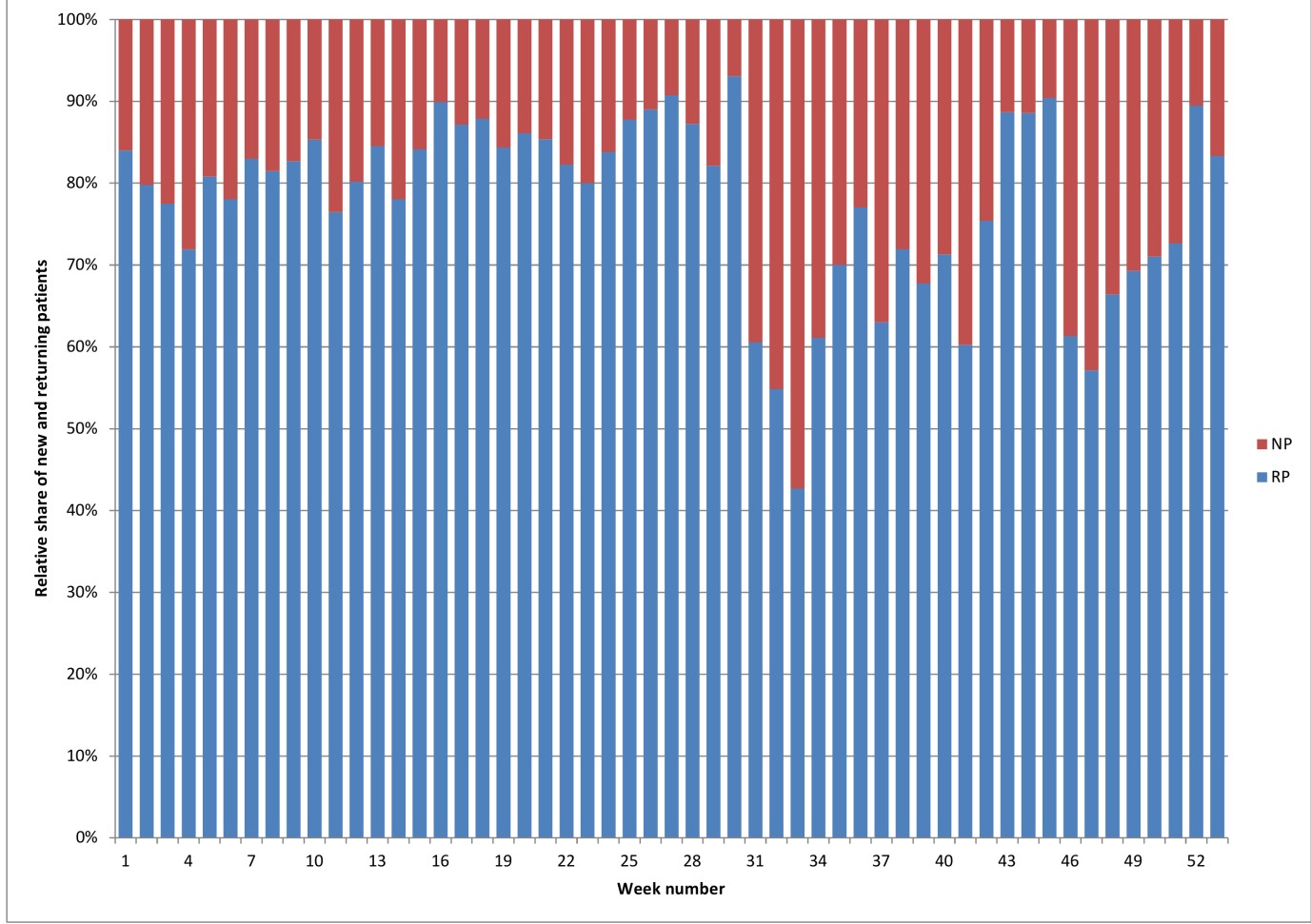

**Figure 5** Relative share of NP and RP per week. NP, new patients; RP, returning patients.

slots became available which allowed the hospital to offer very short access times. As such, the system responded quickly through a market mechanism based on access times. Being aware of the dramatically reduced access times, GPs directly started to refer a greater proportion of their patients to this hospital.

This sudden inflow had consequences for the ratio of new to returning patients, which also changed from week 31 onwards. In the first 30 weeks of the year, there were typically about 13 new patients per week (figure 4) representing 15%–20% of the patients seen (figure 5). However, in weeks 31–49, the average number of new patients increased sharply to an average of 28 per week, while the proportion of new patients correspondingly increased to 30%–40% as of week 31 (figures 4 and 5).

These additional new patients continued to fill the system. However, the increased inflow had a multiplier effect as each new patient admitted would typically have four follow-up appointments in the year following their first visit. In the worst-case scenario, this could result in a situation where there would be insufficient capacity for returning patients.

When we reflect on our findings, the actions of the department seem to have triggered a potential amplification of its backlog. First, by adding capacity, and second by then accepting more new patients than the department would be able to cope with in the longer term. While we do not have accurate database information for the following year, the interviews provided information that new capacity problems did emerge. These capacity problems were a consequence of the vicious cycle that was started by deliberately expanding capacity and occurred despite the best intentions to reduce waiting times and increase patient care quality and satisfaction.

## DISCUSSION

In this case study, we focused on the access times and the capacity management of a hospital's rheumatology department. We first revisit the research question before considering the general implications of our study.

The main research question focused on how patient waiting times vary following an expansion in capacity. Our analysis revealed that the local managerial decision to temporarily add extra capacity to reduce the backlog (ie, shorten access times) had a magnetic effect on GPs who then referred many more patients to this hospital rather than elsewhere. The increase in new patients and, as a consequence, the avalanche of follow-up appointments were disproportionate to the extra capacity.

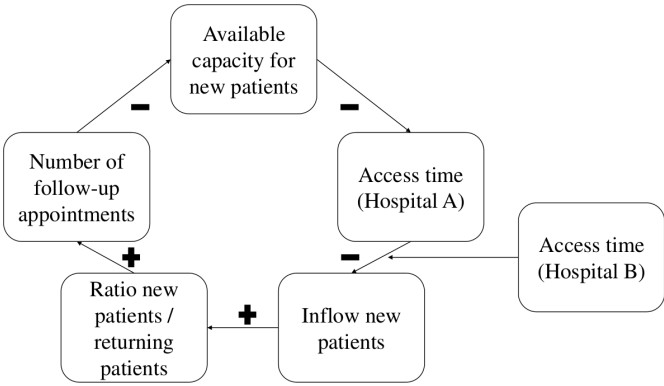

**Figure 6** Causal network of the rheumatology situation.

Furthermore, the decision to allow a significant increase in the inflow of new patients once this new capacity was added resulted, at least in the short term, in a substantial change in the ratio of new to returning patients. With a surge in new patients, most of whom would need four follow-up appointments; it was inevitable that the department would later become overwhelmed with these new patients requiring multiple returning appointments.

Reductions of waiting time result in increased inflow. However, in this case, the temporary aspect of the waiting time reduction affects capacity supply in the short term while creating even more demand for the same capacity in the longer term. A different approach towards waiting time reduction, such as process improvements, would arguably have had a smoother and more structural impact. Also these process improvements could have been expected to lead to increased inflow, but at a level that matched the structural capacity, such that a new equilibrium could be established based on shorter waiting times.

In hindsight, two local decisions, to increase capacity and to use this to accept many more new patients, were important sources of artificial variability which triggered a self-inflicted multiplier effect. This effect was initially considered as a potential example of a service bullwhip effect. The service bullwhip effect also relates to a trigger event in a supply chain that induces demand distortion.[22] However, we argue for a simpler multiplier effect to have taken place. We have constructed a causal network of the rheumatology situation, figure 6, to further explore this multiplier effect.

The network shows that a sharp decrease in the department's access times (here referred to as hospital A) caused GPs to switch their demand by opting to refer patients to hospital A rather than elsewhere (hospital B). This was possible because GPs were aware of the improved access time offered by hospital A, which distinguished it from hospital B. This resulted in a sharp increase in the inflow of new patients of whom an unlimited number were accepted (although one would expect the access times to eventually increase as appointments fill up until a new equilibrium in reached with hospital B). This boosted the ratio of new to returning patients, and resulted in a massive increase in demand for follow-up appointments

due to a multiplier effect with typically four returning visits in the year following first visit. The increase in follow-up appointments absorbed all the available capacity, which had a negative effect on the access times offered to future new patients.

It is striking to see that the increase in new patients has a self-reinforcing effect on the demand for physician capacity due to the multiplier effect of follow-up appointments. This reinforcing effect reveals itself in the next stage of the supply chain and is typical for healthcare environments where follow-ups are common. The increase in inflow is the consequence of continuous and instantaneous information sharing between the hospital (and its rheumatology department) and GPs. Earlier, the GPs' ability to select hospitals based on their access times, acted as a variability absorbing or stabilising mechanism in case of 'normal' variability and gradual changes in wait times. For example, as soon as one of the hospitals would tend to get to a higher inflow of patients and hence increased access times, the GPs responded by selecting different hospitals in the system. In turn, this would bring the inflow level back to the original level, rebalancing the system load. However, in this case, the temporary capacity change disturbed this self-stabilising mechanism. The case stresses that it is insufficient to rely on a local supply chain perspective. Here, system awareness is necessary and this requires considering the referral behaviour of GPs and the delivery performance of other competing hospitals when addressing variability and buffers. The market mechanism, based on access times, that this case study has highlighted, clearly requires a system view to smooth the demand and keep long-term access times at an acceptable level.

The theoretical contributions of this study are threefold. First, we contribute to the knowledge on flow, and consequently the 'theory of swift and even flow'[19] by positioning artificial variability as a managerial problem that negatively affects flow, and we show the possible negative effect of a common buffering approach. The addition of capacity would normally be a means to buffer against day-to-day variability. However, it now became a 'disturbing' factor or source of variability itself for longer-term variability. Based on this study, and given the dynamics of the complex interaction between variability and buffers, we stress the need to apply a system view when pursuing flow improvements. Introducing poorly informed local policies can negatively affect flow.[7]

Second, we contribute to the knowledge base on the role of variability in healthcare environments.[13 14 23] The hospital's well-intended actions caused inflow variability; here our study provides an example of unexpected outcomes related to improvement initiatives. We showed a multiplier effect, one that was self-inflicted and caused by decision-making being based on a local rather than a system perspective. Given the existence of a market mechanism based on access times, supply chain partners need to make decisions based on an overview of available 'market information', and be sensitive to regional effects.

This decision to allow new patients has strong similarities with Roemer's Law,[24 25] which underlines that available capacity will be utilised.

Third, we contribute to the growing knowledge base on process improvement, which includes approaches such as Six Sigma and lean thinking.[26 27] Earlier research identified three paradoxes in healthcare improvement initiatives: (1) improvement initiatives target parts of a system, but fail to address underlying system constraints, (2) local improvements do not account for regional integration and (3) rules that improve the service organisation for one department may create obstacles for other departments.[6] Our results provide illustrations of these paradoxes. In addition, by confirming the importance of variability and buffer reduction, an area that has recently received increasing attention in the literature on lean thinking. Lean is generally considered to include the concept of *mura*, which reflects unevenness or variability as can be found in the healthcare field.[28] Hopp[29] (p.89) considers the 'production of goods or services to be lean if it is accomplished with minimal buffering costs'. After an initial focus on direct waste, more mature lean approaches tend to focus on reducing variability and buffers.[29] This study contributes by providing an example of the interaction between variability and capacity buffers within a hospital setting.

An obvious limitation of this research is that it is based on a single case study, which consequently limits the generalisability of the findings. Another limitation relates to the fact that the service supply chain in this paper is a simple dyadic structure of GPs and a relatively small rheumatology department with new and returning patients. As such, it is not representative of very complex networks with many interdependencies needing to be understood. A strength of the study is its combination of objective quantitative and subjective qualitative data that contributes to the accuracy and reliability of the findings.

We would encourage further research aimed at increasing the knowledge on multiplier effects and the dynamics between variability and buffers. Where this study took a system perspective, a patient's perspective on variability and buffers could add an extra dimension to this field of research.

## CONCLUSIONS

This research has identified the variation in access times that follows from adding capacity. We have seen that access times are heavily influenced by sources of artificial variability such as managerial decisions about adding capacity and accepting new patients. This can cause undesirable dynamics in the variability–buffer interaction. Our findings lead us to agree with earlier research[30] that concluded that synchronising service supply chains is 'complex' and 'non-intuitive'.

**Acknowledgements** The authors would like to thank the health staff at the case site for their cooperation during this research project. The authors thank the reviewers for their constructive comments on earlier versions of this manuscript.

**Contributors** All authors contributed to the design of the research and to carrying it out. PW and FZ collected the data and analysed the results. OR, FZ and KA wrote the manuscript, and PW and ML provided comments on the manuscript and approved the final version.

**Funding** The authors have not declared a specific grant for this research from any funding agency in the public, commercial or not-for-profit sectors.

**Competing interests** None declared.

**Patient consent for publication** Not required.

**Ethics approval** This study was carried out with approval of the ethics committee of the Faculty of Economics and Business, University of Groningen (reference number RDMPFEB-20180111-2835, 23 February 2018).

**Provenance and peer review** Not commissioned; externally peer reviewed.

**Data availability statement** Data are available upon reasonable request.

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
