## [Reviewer comments · BMJ Open]

ARTICLE DETAILS

TITLE (PROVISIONAL)	How improving access times had unforeseen consequences: a case study in a Dutch Hospital
AUTHORS	Roemeling, Oskar; Ahaus, Kees; van Zanten, Folkert; Land, Martin; Wennekes, Patrick

VERSION 1 – REVIEW

REVIEWER	Sara Kreindler University of Manitoba, Canada
REVIEW RETURNED	04-May-2019

GENERAL COMMENTS	In this study, a hospital sought to reduce wait times for rheumatology consultation by adding capacity (staff). This strategy did reduce wait times, but the hospital soon became a victim of its own success, as it was soon inundated by new demand that would have otherwise been referred to other hospitals. The inundation was compounded by the fact that each new patient required several follow-up appointments. These findings show the perils of designing patient flow initiatives locally without a larger system perspective, and they are worth reporting. However, the article has some problems: The study is not well contextualized in the flow literature, the novelty of the findings is overstated, the design is purportedly mixed-methods but the qualitative methods and results are under-reported, and the interpretations in relation to theory are sometimes questionable. 1. Contextualization and novelty. There has been quite a bit of literature on artificial variability in healthcare - e.g., the work of Litvak - but this is not cited. There has also been literature on the tendency of increased healthcare capacity to call forth new demand (e.g., the so-called "woodwork effect," or Roehmer's law that "a bed built is a bed filled), on the reasons why flow interventions often fail ("Six ways not to improve patient flow"), and on the limitations of Lean in complex contexts (e.g., "Complexity complicates lean.") The problem of planning interventions without a system view has also been articulated in other articles, such as "The paradoxes of patient flow," which you cite but not in this regard. I suspect there is also non-healthcare literature on situations in which a service, having improved its wait times, becomes overwhelmed with new or redistributed demand - surely this situation can't be that unusual. There needs to be a thorough account of what is already known and what this study adds. The ways in which the study is novel need to be clearly specified, taking into account past literature; it is clearly not as novel as it claims to be.
---

2. Methods. Although the study is billed as mixed-methods, there is very little detail about the qualitative methods (e.g., how many of what type of people were sampled, on what basis were they selected, what did the interviews vs. other methods contribute, how were the qualitative data analyzed, etc.) or the results (e.g., no data extracts presented). The only contribution that the qualitative methods seem to make is the provision of a piece of factual information: GPs said they were aware of hospital wait times, and quickly responded to the change in Hospital A's wait times by referring more patients there. It might be possible to supply that piece of information even without attempting to report the qualitative component (perhaps citing it as a qualitative component of an overall evaluation of the improvement initiative, not reported here). On the other hand, there may be a missed opportunity in failing to report the qualitative component in greater detail. Did anything else useful come out of the qualitative data - for example, did people offer any insight into why they didn't anticipate that the hospital might be overwhelmed by redirected demand?

The quantitative methods are non-statistical - I didn't really have a problem with that, although you might consider a visual method like statistical process control to test for significance. A minor point: Please report results in chronological order; it is confusing when the paper jumps from 31 weeks to 28 to 46, for instance.

3. Interpretation in relation to theory. This paper states that it is about (a) the unintended consequences of increased capacity, and (b) the introduction of artificial variability. I'm not sure either of these is accurate. So far as we know, the cause of Hospital A's downfall was the fact that it decreased wait times (sparking market signals that attracted clientele from Hospital B), not the fact that it applied a particular wait-reduction strategy (i.e., increased capacity). Any other successful means of reducing wait times, such as process improvement, might have had the same unintended consequence. The fact that the strategy happened to be a capacity increase is only relevant if the presence of additional capacity not merely redirected but distorted demand - i.e., if GPs started to make more liberal referral decisions instead of managing more patients themselves. (It would be nice to rule out that possibility through analysis of patient volumes at Hospital B - was there indeed a decrease in demand at Hospital B proportionate to the increase at Hospital A? But I will understand if data aren't available.) This should be made much clearer in both the paper and the abstract.

As for the introduction of artificial variability, I'm not convinced that the decision to increase a capacity buffer, or the ongoing policy of accepting all eligible referrals, can be considered a source of artificial variability. They don't cause day-to-day variability in flow in the same way as, for example, surgeons' reluctance to work on weekends. It could be argued that GPs' ability to select hospitals on the basis of fluctuations in their wait times is a source of artificial variability. But the more relevant issue seems to be that planners underestimated the scope of the potential pool of demand. In sum, this study really doesn't "reveal the interaction between variability and buffers," nor does it really contribute to the theory of swift, even flow - its contribution is valuable, but more modest.

	The paper also contains a lengthy discussion differentiating this effect from the bullwhip effect. I'm not certain how much this adds, because the effect that was discovered is quite simple and intuitive; it doesn't seem to need to be exhaustively examined in relation to a more complex and subtle process. However, it's true that the study uncovered a different kind of multiplier effect. So if the section is retained, it needs to begin with a clear definition of the bullwhip effect, not just its components but what it actually is - otherwise the reader has to piece this together from bits of information presented throughout the section. Finally, a minor point: The term "admission time" can have different meanings; it may, for instance, refer to the time between presentation at the Emergency Department and admission to an inpatient unit. It wasn't until the middle of the methods section that I discovered how it was defined in this paper. Please either define the term the first time it is used or use a descriptive phrase as an alternative (e.g., "wait time for new patients to see a specialist," later abbreviated to "wait time").
--	---

REVIEWER	Dawn Swancutt University of Plymouth, UK
REVIEW RETURNED	13-May-2019

GENERAL COMMENTS	This paper addresses an interesting area for health service improvement. The paper is clear and well written and will, no doubt, offer an insight for those attempting to improve their own service wait times. There are a number of minor points that would improve the value and readability of the work: Introduction page 3 line 29-31, please add the reference for the statement about the surgeons' late arrival. Methods page 4 line 26, it may be helpful to give an indication of the typical timeline from the GP referral to the treatment plan discussion. Methods page 4 line 30/31, should it read symptoms rather than situations? Methods page 4 line 35-36, can the authors clarify their use of the term admission in the case description section? Is that admission to a bedded ward in the hospital, or attendance at a consultation appointment as an outpatient? Methods page 4, data sources, please confirm that a consent process was carried out with research participants. Page 5, it is disappointing that no public or patient involvement was sought. This might have been easily accommodated in the same way as the focus group meeting with staff, but given a deeper understanding into delays and variability from the patients' perspective. A helpful website explaining how valuable patient involvement can be is found here: https://www.invo.org.uk/. This may offer an avenue of further research interest for the authors. Results page 5 line 44, is the meaning of the intern's capacity realise or should it be release? The 'intern would achieve' may give a clearer meaning to your readers.
--

	Conclusions page 9 line 55, it may be helpful for the reader to qualify that the variation identified was a result of adding capacity in this particular way (i.e. in the way that the new appointments were allocated to the additional staff member). There may be alternative ways for additional staff to support capacity of current patient wait times that create fewer unintended consequences. Could the authors comment on this in the discussion? References – The paper seems to refer to little of the literature on patient flow – it would be helpful to see more comprehensive referencing on patient flow in hospital clinic settings.
--	---

VERSION 1 – AUTHOR RESPONSE

Reviewer 1

Reviewer Name: Sara Kreindler

Institution and Country: University of Manitoba, Canada

Please state any competing interests or state 'None declared': none declared

Please leave your comments for the authors below

In this study, a hospital sought to reduce wait times for rheumatology consultation by adding capacity (staff). This strategy did reduce wait times, but the hospital soon became a victim of its own success, as it was soon inundated by new demand that would have otherwise been referred to other hospitals. The inundation was compounded by the fact that each new patient required several follow-up appointments.

These findings show the perils of designing patient flow initiatives locally without a larger system perspective, and they are worth reporting. However, the article has some problems: The study is not well contextualized in the flow literature, the novelty of the findings is overstated, the design is purportedly mixed-methods but the qualitative methods and results are under-reported, and the interpretations in relation to theory are sometimes questionable.

Response

The reviewer is rightfully pointing out issues in our paper, we have added our in depth responses after the more elaborate reviewer commentary on each of the specific points below. In our responses, we address the following:

1. Contextualization in flow literature;
2. Methods;
3. Interpretation of findings in relation to theory.

1. Contextualization and novelty. There has been quite a bit of literature on artificial variability in healthcare - e.g., the work of Litvak - but this is not cited. There has also been literature on the tendency of increased healthcare capacity to call forth new demand (e.g., the so-called "woodwork effect," or Roehmer's law that "a bed built is a bed filled), on the reasons why flow interventions often fail ("Six ways not to improve patient flow"), and on the limitations of Lean in complex contexts (e.g., "Complexity complicates lean.") The problem of planning interventions without a system view has also been articulated in other articles, such as "The paradoxes of patient flow," which you cite but not in this regard. I suspect there is also non-healthcare literature on situations in which a service, having improved its wait times, becomes overwhelmed with new or redistributed demand - surely this situation can't be that unusual. There needs to be a thorough account of what is already known and

what this study adds. The ways in which the study is novel need to be clearly specified, taking into account past literature; it is clearly not as novel as it claims to be.

Response

The reviewer addresses the issue on contextualization, and especially embedding the paper in the flow literature. Following the reviewer's suggestions, we have expanded the introduction and discussion to contain more information on variability in healthcare, and better position our findings. We feel particularly sorry for having missed the articles of the reviewer on "Paradoxes of patient flow" (Kreindler, 2017a) and "Six ways not to improve patient flow" (Kreindler, 2017b) as this was definitely relevant in the light of our article. Besides these, we have now included works of several authors, including Fredendall et al. (2009), Sloan et al. (2014), and Litvak et al. (2005). In addition, we now link our paper more explicitly to the Theory of Swift and Even Flow and touch upon the Theory of Constraints.

We now refer to Roemer's Law as a related phenomenon. Yet, we are not entirely sure the woodwork effect is the exact phenomenon we witness in our case. The woodwork effect relates strongly to policy aspects of the availability of care, where increased availability (i.e. insurer coverage) will entice patients to come from 'the woodwork' and use the offered services or enroll for coverage. In The Netherlands, all inhabitants have a compulsory health insurance, and we could not identify any major healthcare reforms that could have influenced the demand for rheumatology care. Roemer's Law relates to availability of hospital beds, which does provide a clear link with capacity availability in our study. For both the woodwork effect as well as Roemer's Law, the overall demand for care appears to increase as a consequence of increased availability or capacity. However, in our research, as far as we could know, the total demand per capita did not increase. The referral strategy of GPs appears a main cause for the increased inflow, and it seems it did not induce new demand. In other words, the increased inflow into the rheumatology department might not have consisted of patients in the woodworks, these were already present albeit waiting at a different care site; patients come from a different part of the system which strongly relates to the regional influence discussed by Kreindler (2017a). As a consequence of these changes, we have (almost completely) rewritten the introduction.

As a consequence of our extended literature discussion, we have toned-down the statements related to our novel contribution and now take a more modest and probably more realistic perspective.

2. Methods. Although the study is billed as mixed-methods, there is very little detail about the qualitative methods (e.g., how many of what type of people were sampled, on what basis were they selected, what did the interviews vs. other methods contribute, how were the qualitative data analyzed, etc.) or the results (e.g., no data extracts presented). The only contribution that the qualitative methods seem to make is the provision of a piece of factual information: GPs said they were aware of hospital wait times, and quickly responded to the change in Hospital A's wait times by referring more patients there. It might be possible to supply that piece of information even without attempting to report the qualitative component (perhaps citing it as a qualitative component of an overall evaluation of the improvement initiative, not reported here). On the other hand, there may be a missed opportunity in failing to report the qualitative component in greater detail. Did anything else useful come out of the qualitative data - for example, did people offer any insight into why they didn't anticipate that the hospital might be overwhelmed by redirected demand?

The quantitative methods are non-statistical - I didn't really have a problem with that, although you might consider a visual method like statistical process control to test for significance. A minor point: Please report results in chronological order; it is confusing when the paper jumps from 31 weeks to 28 to 46, for instance.

Response

The reviewer points out an issue that was a struggle for us: the role of the qualitative data. Our study is mostly quantitative in nature and we rely on the quantitative data set to show the development of throughput time and waiting time for patients in a rheumatology department. However, whilst the quantitative data is obviously able to show trends, and provides an insight into performance, it does not explain what happened.

During meetings with department representatives, we discussed our findings and tried to understand why and how performance changed over time. Whilst obviously qualitative in nature, the meetings were less structured than a regular series of interviews. In hindsight, this might be a missed opportunity. Yet, we felt that omitting the qualitative part would not be a rightful representation of the research process and opted for the mixed-methods terminology. This being said, we take the reviewers' comments to heart and follow the advice provided here. We have changed the text to "a quantitative study with a qualitative component". Considering the limited role of the qualitative data in our study, this description appears more adequate.

We have taken a new look at our presentation of our findings, and have added t-tests to compare between two (seemingly) distinct periods; before and after the changes in capacity. Whilst a modest change, it does provide some statistical backing to our main findings. We have made small changes to improve the chronological order in the results. For each of the graph we now strictly follow the timeline. Only for Figure 1 we made the choice to first discuss all the main issues in order of time, before we discuss two remarkable notions that happened in some specific weeks. If the reviewer prefers differently, we are willing to change this too, but we have the impression this would just increase confusion.

3. Interpretation in relation to theory. This paper states that it is about (a) the unintended consequences of increased capacity, and (b) the introduction of artificial variability. I'm not sure either of these is accurate. So far as we know, the cause of Hospital A's downfall was the fact that it decreased wait times (sparking market signals that attracted clientele from Hospital B), not the fact that it applied a particular wait-reduction strategy (i.e., increased capacity). Any other successful means of reducing wait times, such as process improvement, might have had the same unintended consequence. The fact that the strategy happened to be a capacity increase is only relevant if the presence of additional capacity not merely redirected but distorted demand - i.e., if GPs started to make more liberal referral decisions instead of managing more patients themselves. (It would be nice to rule out that possibility through analysis of patient volumes at Hospital B - was there indeed a decrease in demand at Hospital B proportionate to the increase at Hospital A? But I will understand if data aren't available.) This should be made much clearer in both the paper and the abstract.

Response

We agree with the reviewer that, in essence, any reduction of waiting time would have resulted in increased inflow. We do believe that the chosen strategy, a temporary addition of capacity, is of importance, as it induced redirection of demand. The temporary aspect implies that it only affects capacity supply in the short term, while creating even more demand for the same capacity in the longer term. Process improvements would have a more smooth and structural impact. Also process improvements could have been expected to lead to increased inflow, but at a level that matched the structural capacity, such that a new equilibrium could be established based on shorter waiting times. Hence, we would argue that such an approach would prove to be more robust. As the reviewer already assumed, unfortunately we only have access to the data of this hospital. Our assumption that volumes for the two hospitals together would not change is fully based on the market knowledge of the staff of the rheumatology department.

In the paper, we have added a shortened deliberation on this issue to the discussion section, it reads as follows:

“Reductions of waiting time result in increased inflow. However, in this case the temporary aspect of the waiting time reduction affects capacity supply in the short term, while creating even more demand for the same capacity in the longer term. A different approach towards waiting time reduction, such as process improvements, would arguably have had a smoother and more structural impact. Also these process improvements could have been expected to lead to increased inflow, but at a level that matched the structural capacity, such that a new equilibrium could be established based on shorter waiting times.

As for the introduction of artificial variability, I'm not convinced that the decision to increase a capacity buffer, or the ongoing policy of accepting all eligible referrals, can be considered a source of artificial variability. They don't cause day-to-day variability in flow in the same way as, for example, surgeons' reluctance to work on weekends. It could be argued that GPs' ability to select hospitals on the basis of fluctuations in their wait times is a source of artificial variability. But the more relevant issue seems to be that planners underestimated the scope of the potential pool of demand. In sum, this study really doesn't "reveal the interaction between variability and buffers," nor does it really contribute to the theory of swift, even flow - its contribution is valuable, but more modest.

Response

The first moment we refer to artificial variability in our results is when stating that there were more weeks in which at least one rheumatologist was partly absent than weeks in which both were fully present. This relates to the day-to-day variability that the reviewer refers to. Indeed “the decisions to increase capacity and to use this to accept many more new patients (including their follow-up visits)” are one time decisions. However these decisions created a “shock” in the system that brings it out of its normal “equilibrium”. We felt that these kinds of shocks, as they are created by the decision makers, could also be seen as artificial variability introduced in the system. After all, Litvak and Long (2000, p. 309) describe artificial variability ‘as flow or professional variabilities caused by a dysfunctional process within the healthcare delivery system.’ According to this meaning of artificial variability the decision to increase capacity or to accept all eligible referrals of patients (including their follow-up visits) can be regarded as dysfunctional process in the system or as variability caused by an ill-thought-out decision. We agree with the reviewer that this is different from day-to-day variability. But, while introducing extra capacity would normally be a means to buffer against this day-to-day variability in inflow it now became a ‘disturbing’ factor or source of variability itself for longer term variability. This is what we meant by interaction between variability and buffers. We have been more precise in explaining this in the discussion part of the paper, which hopefully resolves the problem.

The GPs' ability to select hospitals based on their wait times, acted as a variability absorbing or stabilizing mechanism in case of ‘normal’ variability and gradual changes in wait times. As soon as one of the hospitals would tend to get to a higher inflow level, with increasing wait times, the GPs' response would bring the inflow level back to the original level. However, the temporary capacity change disturbed this stabilizing mechanism. We have adapted our discussion accordingly.

The paper also contains a lengthy discussion differentiating this effect from the bullwhip effect. I'm not certain how much this adds, because the effect that was discovered is quite simple and intuitive; it doesn't seem to need to be exhaustively examined in relation to a more complex and subtle process. However, it's true that the study uncovered a different kind of multiplier effect. So if the section is retained, it needs to begin with a clear definition of the bullwhip effect, not just its components but what it actually is - otherwise the reader has to piece this together from bits of information presented throughout the section.

Response

When we were first confronted with the multiplier effect in the data, we thought the phenomenon was related to the service bullwhip effect as identified by Akkermans and Voss (2013). However, when we developed our thinking over time, we became aware of the differences. We replaced this lengthy discussion by a much shorter reference to the Bullwhip effect, and then continue with a condensed explanation of the multiplier effect.

Finally, a minor point: The term "admission time" can have different meanings; it may, for instance, refer to the time between presentation at the Emergency Department and admission to an inpatient unit. It wasn't until the middle of the methods section that I discovered how it was defined in this paper. Please either define the term the first time it is used or use a descriptive phrase as an alternative (e.g., "wait time for new patients to see a specialist," later abbreviated to "wait time").

Response

Thank you for making us aware of this. We made an error in directly translating the term used in Dutch hospitals to English, without realising the different meaning in English. We have now adopted the term access time and explicitly define it in the text at the start of the introduction as: "the time between referral from the GP to the first visit of the patient to the rheumatology department".

We thank the first reviewer for the insightful comments and helpful suggestions. We hope that our adaptations of our manuscript do justice to your inputs, and that you are satisfied with our changes.

Reviewer: 2

Reviewer Name: Dawn Swancutt

Institution and Country: University of Plymouth, UK

Please state any competing interests or state 'None declared': None declared

Please leave your comments for the authors below

This paper addresses an interesting area for health service improvement. The paper is clear and well written and will, no doubt, offer an insight for those attempting to improve their own service wait times. There are a number of minor points that would improve the value and readability of the work:

Response

We are pleased to read that the reviewer's overall response is positive. The reviewers' comments have definitely helped us to improve the current manuscript.

Introduction page 3 line 29-31, please add the reference for the statement about the surgeons' late arrival.

Response

We have added a reference to the study of Dempsey (2009) which clearly shows examples of artificial variability.

Methods page 4 line 26, it may be helpful to give an indication of the typical timeline from the GP referral to the treatment plan discussion.

We have added a short statement to indicate the typical timeline for a patient that moves from GP to rheumatology department.

Methods page 4 line 30/31, should it read symptoms rather than situations?

The reviewer is correct in stating that the text should read 'symptoms', and not 'situations'. This has been corrected.

Methods page 4 line 35-36, can the authors clarify their use of the term admission in the case description section? Is that admission to a bedded ward in the hospital, or attendance at a consultation appointment as an outpatient?

Thanks to the reviewer comments, we are now aware of the difference in admission time terminology. We made an error in directly translating the term used in Dutch hospitals to English, without realising the different meaning in English. We have now opted for a different term access time, and explicitly define it in the first paragraph of the introduction, it reads "the time between referral from the GP to the first visit of the patient to the rheumatology department".

Methods page 4, data sources, please confirm that a consent process was carried out with research participants.

Regarding the informed consent, we have added the explicit statement to indicate that all respondents were aware of their rights during the research, and participated of their own free will.

Page 5, it is disappointing that no public or patient involvement was sought. This might have been easily accommodated in the same way as the focus group meeting with staff, but given a deeper understanding into delays and variability from the patients' perspective. A helpful website explaining how valuable patient involvement can be is found here: <https://www.invo.org.uk/>. This may offer an avenue of further research interest for the authors.

Response

We agree that including patient perspective into this research could have been very valuable. Our current study is based upon a completed improvement project, hence the opportunity to include patient perspectives had passed. Nevertheless, we take the advice of the reviewer to heart, and will explore the possible role of patient input for future studies. In addition, we have added a short related statement to the limitations of our study.

Results page 5 line 44, is the meaning of the intern's capacity realise or should it be release? The 'intern would achieve' may give a clearer meaning to your readers.

Response

We gladly follow the useful suggestion of the reviewer. We have altered the text to read:

"Here, we note that the productivity of the resident was expected to be similar to that of the rheumatologists, whereas the intern would achieve roughly one third of their output due to a lack of experience. The graph is adjusted to reflect this difference. In effect, by increasing staff, the department sought to replace time buffers (waiting patients) with capacity buffers (additional physicians).".

Conclusions page 9 line 55, it may be helpful for the reader to qualify that the variation identified was a result of adding capacity in this particular way (i.e. in the way that the new appoints were allocated to the additional staff member). There may be alternative ways for additional staff to support capacity of current patient wait times that create fewer unintended consequences. Could the authors comment on this in the discussion?

Response

We agree and have now discussed that part of the problem relates to the fact that the extra capacity has been used for accepting more new patients, while preferably it should have only been used for new patients that would have been referred without the capacity change. The problem in the Dutch

situation is that the requirements for transparency on waiting times will always reveal the availability of short term capacity (with available slots being visible for the GP), while the hospital cannot refuse patients.

References – The paper seems to refer to little of the literature on patient flow – it would be helpful to see more comprehensive referencing on patient flow in hospital clinic settings.

Response

The reviewer has a good point, the original manuscript did not reflect sufficiently on flow and the role of variability. We have made changes to our introduction and discussion sections to include a more extensive overview of flow related literature. We have added references to McManus et al. (2003), Litvak et al. (2005), Ryckman et al. (2009), Delamater et al. (2013), Bard et al. (2016), Kreindler (2017) and Potts et al. (2018) to better embed this paper in the literature on flow.

We thank the second reviewer for the thorough reading of our paper and helpful comments. We tried to incorporate all suggestions and hope that our changes to the manuscript are satisfactory.

VERSION 2 – REVIEW

REVIEWER	Sara Kreindler University of Manitoba, Canada
REVIEW RETURNED	31-Jul-2019

GENERAL COMMENTS	The authors have addressed the review comments in a thorough and thoughtful way. A minor correction that should not require re-review: The p-value should be reported as $p < .001$ or $<.0001$ (.000 implies zero probability that the effects were due to chance, which cannot be proven).
---

REVIEWER	Dawn Swancutt University of Plymouth, UK
REVIEW RETURNED	30-Jul-2019

GENERAL COMMENTS	The paper is much improved by the authors modifications.
--